



# A large-sample modelling approach towards integrating streamflow and evaporation data for the Spanish catchments

Patricio Yeste[1,2,3], Matilde García-Valdecasas Ojeda[2,3], Sonia R. Gámiz-Fortis[2,3], Yolanda Castro-Díez[2,3], Axel Bronstert[1], and María Jesús Esteban-Parra[2,3]

[1]Institute of Environmental Science and Geography, University of Potsdam, Karl-Liebknecht-Straße 24–25, 14476 Potsdam, Germany
[2]Department of Applied Physics, University of Granada, Campus Fuentenueva S/N, 18071 Granada, Spain
[3]Andalusian Institute for Earth System Research (IISTA-CEAMA), University of Granada, 18006 Granada, Spain

**Correspondence:** Patricio Yeste (patricio.yeste@uni-potsdam.de)

**Abstract.** The simultaneous incorporation of streamflow and evaporation data into sensitivity analysis and calibration approaches has a great potential to improve the representation of hydrologic processes in modelling frameworks. This work aims to investigate the capabilities of the Variable Infiltration Capacity (VIC) model in a large-sample application focused on the joint integration of streamflow and evaporation data for 189 headwater catchments located in Spain. The study has been articulated into three parts: (1) a regional sensitivity analysis for a total of 20 soil, routing and vegetation parameters to select the most important parameters conducive to an adequate representation of the streamflow and evaporation dynamics; (2) a two-fold calibration approach against daily streamflow and monthly evaporation data based on the previous parameter selection for VIC, and (3) an evaluation of model performance based on a benchmark comparison against a well-established hydrologic model for the Spanish domain and a cross-validation test using multiple meteorological datasets to assess the generalizability of the calibrated parameters. The regional sensitivity analysis revealed that only two vegetation parameters – namely, the leaf area index and the minimum stomatal resistance – were sufficient to improve the performance of VIC for evaporation. These parameters were added to the soil and routing parameter during the calibration stage. Results from the two calibration experiments suggested that, while the streamflow performance remained close in both cases, the evaporation performance was highly improved if the objectives for streamflow and evaporation were combined into a single composite function during optimization. The VIC model outperfomed the reference benchmark and the independent meteorological datasets yielded a slight to moderate loss in model performance depending on the calibration experiment considered. This investigation will help gain a better understanding of the hydrology of the Spanish catchments and will help prepare the ground for a fully gridded implementation of the VIC model in Spain.

## 1 Introduction

Large-sample hydrology (Addor et al., 2020) and large-scale hydrology (e.g., Bierkens, 2015; Wood et al., 2011) aim to promote the transferability of knowledge between regions and assess the applicability of hydrologic models and theories at regional, continental and global scales. Large-sample hydrology involves large sets (tens to thousands) of catchments, and its





main focus is to provide generalizable knowledge of hydrological processes and models based on a large sample of catchments representing different hydroclimatic conditions (Addor et al., 2020). Similarly, large-scale hydrology relies on simulations from land-surface models carried out at the so-called spatial hyper-resolution (> 1 km) to quantify and monitor the terrestrial water cycle at multiple scales (Bierkens, 2015; Bierkens et al., 2015; Wood et al., 2011).

Large-sample and large-scale hydrologic studies play an important role in supporting water resources planning and quantifying hydrologic changes across scales in the context of a changing climate (Addor et al., 2020; Wood et al., 2011). The gap between both hydrologic fields is becoming increasingly reduced, and both can be considered as two complementary approaches that attempt to provide a solid understanding of the spatial variability of hydrologic processes and to facilitate the intercomparison of model structures across climates (Addor et al., 2020). This is also manifested in the greater areas covered by hydrologic models (Beck et al., 2016, 2020), the development of gridded runoff observations (Gudmundsson and Seneviratne, 2016; Ghiggi et al., 2019), the tendency towards finer resolutions in land-surface models (Bierkens, 2015; Wood et al., 2011), and the use of macroscale hydrologic models in large-sample studies (e.g., Mizukami et al., 2017; Newman et al., 2017; Rakovec et al., 2016a, b, 2019; Sepúlveda et al., 2022; Yeste et al., 2020, 2021).

From a data perspective, large-sample hydrology encompasses hydrologic studies founded on large-sample datasets of streamflow observations, hydrometeorological data and hydroclimatic and landscape attributes (Addor et al., 2020; Kratzert et al., 2023). This includes investigations on extreme events (e.g., Blöschl et al., 2017; Do et al., 2017; Gudmundsson et al., 2019), climate change impacts (e.g., Marx et al., 2018; Melsen et al., 2018; Vormoor et al., 2015), variations in terrestrial water storage (e.g., Zhang et al., 2017), model evaluation and benchmarking (e.g., Aerts et al., 2022; Newman et al., 2017; Rakovec et al., 2019; Yeste et al., 2020), data and modelling uncertainties (e.g., Beck et al., 2017; Coxon et al., 2015), parameter estimates during calibration (e.g., Beck et al., 2016, 2020; Mizukami et al., 2017; Rakovec et al., 2016a, b), and transferability of parameters in space based on parameter regionalization techniques (e.g., Beck et al., 2020; Pool et al., 2021; Rakovec et al., 2019). But over and above the extensive hydroclimatic characterization commonly provided in large-sample datasets, streamflow is considered a category of its own (Addor et al., 2020). Streamflow datasets are primarily based on individual contributions from national hydrologic services, which constitute the building blocks of continental and global streamflow repositories. The role of national water archives is of capital importance in this respect, and ultimately, it is the international collaboration among national authorities worldwide which makes it possible to tackle this complex challenge (Addor et al., 2020).

Large-sample hydrologic studies can strongly benefit from the integration of satellite remote sensing data into modelling frameworks in order to draw more robust conclusions on catchment functioning (Clark et al., 2017; Rakovec et al., 2016a, b, 2019; Yeste et al., 2020, 2021, 2023). In particular, the use of satellite-based algorithms to retrieve evaporation information represents an unprecedented opportunity to monitor the dynamics and the climate-driven changes in evaporative fluxes (Konapala et al., 2020; Koppa et al., 2022). Evaporation represents the second largest component of the global water balance and is expected to increase as a consequence of global warming (IPCC, 2021). These changes can pose a challenge for future water security and water resources availability from regional to global scale (Lehner et al., 2019; Konapala et al., 2020; Koppa et al., 2022). Therefore, the integration of evaporation data into large-sample modelling approaches is a promising solution to calibrate and





evaluate models for more than one hydrologic variable (traditionally streamflow, Dembélé et al., 2020a, b) and thus achieve a more reliable quantification of the water balance (Yeste et al., 2023).

This study aims to develop a hydrologic modelling framework to investigate the streamflow and evaporation dynamics for a large set of Spanish catchments. As part of the Iberian Peninsula, Spain constitutes a region where the effects of climate change are already noticeable and are expected to be much more pronounced by the end of the 21th century (IPCC, 2021). The Iberian Peninsula has been previously identified as a hotspot (Diffenbaugh and Giorgi, 2012; Vogel et al., 2021), and has manifested recurrent droughts and an increasing tendency towards aridity conditions for the last decades (García-Valdecasas Ojeda et al.,

2021a, b; Páscoa et al., 2017). From a hydrologic perspective, the Spanish catchments have undergone dramatic streamflow decreases during the last decades (Lorenzo-Lacruz et al., 2012, 2013), and evaporative fluxes play a dominant role in the water balance for the entire region (García-Valdecasas Ojeda et al., 2020a; Vicente-Serrano et al., 2014). These changes are expected to exacerbate under climate change (García-Valdecasas Ojeda et al., 2020b, 2021a, b) and can pose an important threat for the future water planning and management in the country.

The large-sample modelling approach followed in this work will be focused on the Variable Infiltration Capacity (VIC) model (Liang et al., 1994, 1996), one of the most widely used hydrologic models in hydrologic studies (Addor and Melsen, 2019). The VIC model has been successfully implemented in many previous large-sample studies and large-scale applications (e.g., Melsen et al., 2018; Mizukami et al., 2017; Rakovec et al., 2019; Sepúlveda et al., 2022), including in Spain (Yeste et al., 2020, 2021), which makes it an excellent choice for the purpose of this investigation. The capabilities of VIC to integrate streamflow

observations and satellite-based evaporation data will be thoroughly examined and its performance will be compared against current modelling efforts providing the basis for water resource planning and management in Spain.

## 2    Study area and data

### 2.1    The Spanish catchments and streamflow dataset

This work is focused on a set of 189 headwater catchments defined for 94 reservoirs and 95 gauging stations belonging to the

main River Basin Districts in Spain (Fig. 1a, Table 1). These catchments are representative of the hydroclimatic variability within the country, and their physiography comprises areas ranging from 9 to 3825 km$^2$, mean elevations from 147 to 1982 m and mean slopes from 4 to 100 m/km (see also Fig. 3 and Table 3, which will be later introduced in Section 3.2). Streamflow observations for the Spanish catchments are monitored in Automatic Hydrological Information Systems (SAIHs, Sistemas Automáticos de Información Hidrológica) and in the Official Network of Gauging Stations (ROEA, Red Oficial de Estaciones

de Aforo) and are estimated via a daily water balance of water storages and releases for reservoirs and using rating curves for gauging stations.

The 189 study catchments were selected from the Integrated Network of Gauging Stations (SAIH-ROEA) dataset (https://www.miteco.gob.es/en/cartografia-y-sig/ide/descargas/agua/anuario-de-aforos.aspx), a national archive of streamflow observations maintained and annually updated by the Spanish Center for Public Work Experimentation and Study (CEDEX, Centro

de Estudios y Experimentación de Obras Públicas). Similarly to Yeste et al. (2018, 2020, 2023), the study catchments were





selected considering a maximum percentage of missing values in the streamflow series of 10% for the period Oct 1990 - Sep 2010, which was chosen as the study period in this work. Among those catchments, 171 (84 reservoirs + 87 gauging stations) presented less than 5% of missing values, and 144 (80 reservoirs + 64 gauging stations) were below 1%.

An exploratory data analysis of negative values during the study period was subsequently conducted and revealed that 47
reservoirs presented up to a 5% of negative estimates of daily streamflow and 46 reservoirs more than 5%, whereas all the 95 gauging stations and only 1 reservoir did not present values below 0 (Fig. 2a). In addition, the percentage ratio of negative to positive values was calculated for each reservoir to quantify their relative importance, suggesting that negative values are close to 0 for reservoirs with less than 5% of negative records and become more visible above 5% (the median percentage ratio of negative to positive values for reservoirs with more than 5% of negative records is 2.9%).

One feasible explanation for the presence of negative values in the streamflow series of reservoirs in the SAIH-ROEA dataset is that inflow data are calculated applying a daily water balance exclusively to water storages and releases. As opposed to gauging stations, where streamflow is derived from rating curves, the daily water balance in reservoirs can produce negative estimates of streamflow (i.e. inflow) when the variation in the storage is negative and its magnitude is greater than the water releases. Given the prominent role of evaporative fluxes in the Spanish catchments, the presence of negative values is likely
to happen in the warmest days of the hydrologic year, when streamflow is minimum or null and open water evaporation in reservoirs become relevant. To further test this hypothesis, the monthly distribution of negative values for all the catchments was calculated and compared against the average hydrologic year of streamflow (Fig. 2b). Results confirm that most negative values occur in summer months, emerging in late spring and extending to the beginning of the hydrologic year. The hypothesis is also supported by the location of reservoirs with more than 5% of negative records as they are mostly concentrated in
Southern Spain (Fig. 2a) and are thus characterized by a warmer climate.

Unfortunately, incorporating the effect of evaporative fluxes into the water balance requires additional data that are not provided in the SAIH-ROEA dataset and that are neither publicly disclosed nor available for all the reservoirs, such as pan evaporation measurements and elevation-area-capacity curves. Hence, this course of action could not be adopted and was left out of the scope of this work. The effect of other potential driving factors for the negative records such as seepage losses
is supposed to be minor in comparison to open water evaporation as negative estimates tend to happen in summer for the southernmost reservoirs. Therefore, on the basis of this initial exploratory analysis, negative values were considered as null and all the 189 headwater catchments were included in the modelling framework. The validity of such assumption will be further discussed in the light of results from the modelling exercise.

## 2.2 Meteorological forcings and evaporation dataset

Daily precipitation and temperature data were collected from the Spanish PREcipitation At Daily scale (SPREAD, Serrano-Notivoli et al., 2017) and the Spanish TEmperature At Daily scale (STEAD, Serrano-Notivoli et al., 2019) datasets, two gridded products at ∼ 5 km resolution constructed by interpolating daily observations from a dense network of meteorological stations distributed across Spain. Monthly evaporation data were collected at 0.25º resolution from the Global Land Evaporation Amsterdam Model (GLEAM) version 3.5a (Martens et al., 2017; Miralles et al., 2011) and were remapped to the study catchments



following a first-order conservative approach. GLEAM has shown less uncertainty compared to other satellite-based evaporation products (Xu et al., 2019) and has found extensive use within hydrologic studies for calibrating and evaluating of hydrologic models (e.g., Bouaziz et al., 2021; Dembélé et al., 2020a, b; Koppa et al., 2019; Mei et al., 2023), particularly in studies involving data-scarce areas (e.g., Dembélé et al., 2020a, b; López López et al., 2017) and/or regions where evaporative fluxes are dominant (e.g., Dembélé et al., 2020a, b; Yeste et al., 2020, 2021, 2023).

Fig. 1b and Fig. 1c show the computed values of the runoff ratio ($Q/P$) and the sum of the runoff and evaporation ratio to precipitation ($(Q+E)/P$) for the study period and the hydroclimatic datasets described in this and the previous section. Two thirds of the catchments manifested $Q/P$ estimates below 0.4 and were predominantly located in the southeastern sector of the country (Fig. 1b). Approximately 50% of the catchments produced $(Q+E)/P$ estimates between 0.9 and 1.1, with negative imbalances ($(Q+E)/P < 1$) mostly corresponding to catchments with a low runoff ratio and positive imbalances

($(Q+E)/P > 1$) towards the northwest (Fig. 1c). The effect of these imbalances will be thoroughly examined in the light of results.

## 3 Methods

### 3.1 The VIC model

The Variable Infiltration Capacity (VIC) model (Liang et al., 1994, 1996) version 4.2.d is a semi-distributed macroscale hy-
drologic model implementing the water and energy balances within a gridded domain at daily and subdaily time steps. The model utilizes a three-layer soil profile to conceptualize runoff generation. Surface runoff, based on the Xinanjiang formulation (Zhao et al., 1980), occurs in the first two soil layers, while baseflow is generated in the bottom layer using the Arno equation (Franchini and Pacciani, 1991). The VIC model considers the subgrid variability in land uses through vegetation tiles. Evaporation in each grid cell is calculated as the sum of evaporation from bare soil, evaporation from the canopy interception and
transpiration, and is constrained by the atmospheric demand for water vapor according to the Penman-Monteith equation. The model structure includes a snow model for accumulation and melting processes, employing snow bands to consider subgrid variability in topography, land uses, and precipitation that makes it applicable across diverse geographical domains. While originally designed as a land surface scheme for Earth system models, VIC has found extensive use globally as a hydrologic model and stands out for its widespread usage within the hydrologic community (Addor and Melsen, 2019).

The VIC model was applied with a gridded configuration at 0.05° resolution ($\sim$ 5 km) and choosing a spin-up period of 10 years preceding the study period. Meteorological forcings were interpolated to the model resolution through a nearest neighbor assignment. Notably, the VIC model lacks consideration for horizontal fluxes between adjacent grid cells, typically addressed by coupling a routing model. A gamma function was selected in this study to post-process the runoff simulations and account for the delay between runoff generation and catchment discharge (i.e., streamflow). The required soil and vegetation
parameters to run VIC were collected at 1 km resolution from the following datasets: bulk density and soil textural classes from SoilGrids1km (Hengl et al., 2014); porosity, saturated hydraulic conductivity, field capacity, and wilting point from EU-SoilHydroGrids ver1.0 (Tóth et al., 2017); land uses from the UMD Global Land Cover Classification (Hansen et al.,





2000), with associated vegetation parameters aligned with Global Land Data Assimilation System (GLDAS) specifications for VIC (Rodell et al., 2004). Soil parameters were regridded to the model resolution using a first-order conservative remapping,
whereas land uses were kept at their original resolution as the subgrid variability in land uses is handled statistically within VIC.

## 3.2 Regional sensitivity analysis

Parameter sensitivities were analysed using the implementation of the Regional Sensitivity Analysis (RSA) method of Hornberger and Spear (1981) in the SAFE Toolbox (Pianosi et al., 2015). RSA is based on a classification of model simulations into
behavioral and non-behavioral according to one or more performance metrics and evaluates differences between parameter distributions corresponding to both classes. The RSA sensitivity index for a given parameter represents the maximum vertical distance between the Cumulative Distribution Functions (CDFs) corresponding to the behavioral and non-behavioral classes, which is equivalent to the Kolmogorov-Smirnov distance statistic computed in the Kolmogorov-Smirnov test. Hence, the RSA sensitivity index ranges from 0 to 1, with values closer to 1 indicating a greater parameter sensitivity.
For each of the 189 catchments, the parametric space was explored conducting a Monte Carlo simulation for 10,000 Latin Hypercube samples (Iman and Conover, 1982) extracted from the parameter ranges of the 20 soil, vegetation and routing parameters analysed in Yeste et al. (2023) and that are described in Table 2. RSA was applied to the Nash-Sutcliffe Efficiencies for the Monte Carlo simulations of daily streamflow ($NSE(Q_d)$) and monthly evaporation ($NSE(E_m)$) calculated for the study period, choosing the median $NSE(Q_d)$ and median $NSE(E_m)$ to classify behavioral and non-behavioral simulations.
Similarly to Sepúlveda et al. (2022), the Spearman correlation coefficient ($r_S$) between the RSA sensitivity indices for $NSE(Q_d)$ and $NSE(E_m)$ and the physiographic and hydroclimatic characteristics defined in Table 3 and depicted in Fig. 3 was calculated. $r_S$ measures how similar the spatial patterns of the parameter sensitivities and the selected attributes are, their sign being indicative of a matching pattern (i.e., positive sign) or an opposite pattern (i.e., negative sign). These attributes were initially selected based on their ease of access for the study catchments and allowed for further investigating the potential
drivers of parameter sensitivities.

The two most influential vegetation parameters to any of the two performance metrics were finally selected for each catchment and incorporated together with the five soil parameters and the two routing parameters (SR parameterization hereafter) into the calibration stage. Adding two extra VIC vegetation parameters to the SR parameterization is sufficient to improve the joint performance against streamflow and evaporation data according to previous research in Yeste et al. (2023).

## 3.3 Calibration and evaluation approach

A Split-Sample Test (SST, Klemeš, 1986) was applied to calibrate and evaluate VIC considering two independent periods of equal duration belonging to the study period: a calibration period from Oct 1990 to Sep 2000 and a evaluation period from Oct 2000 to Sep 2010. A spin-up simulation of 10 hydrologic years preceding both the calibration and evaluation periods was conducted to provide initial states of model storages free from the effect of initial conditions (this strategy was also applied



to the Monte Carlo simulation described in the previous section). The performance of the VIC model was evaluated through $NSE(Q_d)$ and $NSE(E_m)$ and their decomposition into correlation $r$, variability $\alpha$ and bias $\beta$.

The calibration was performed using the Shuffled-Complex-Evolution Algorithm (SCE-UA) of Duan et al. (1994) and following a single-objective optimization approach for the nine selected parameters (five soil parameters, two routing parameters and two vegetation parameters) to minimize a composite function that aggregates the performance metrics for streamflow and evaporation:

$$\text{Minimize} \quad \sqrt{w_Q \cdot (1 - NSE(Q_d))^2 + w_E \cdot (1 - NSE(E_m))^2} \tag{1}$$

This problem minimizes the two-dimensional weighted Euclidean to the ideal vector (1,1) and belongs to the more general weighted-metric method to minimize distances (see Yeste et al., 2023, for a detailed implementation of this problem to integrate streamflow and evaporation data). In this work, two calibration experiments for different weights combinations in Eq. 1 were applied to the VIC model: firstly, a streamflow-only calibration (Q-only calibration hereafter) was performed by choosing $w_Q = 1$ and $w_E = 0$. Secondly, the model was calibrated for a weighted Euclidean distance (Q-E calibration hereafter) selecting two equal weights $w_Q = w_E = 0.5$. The case of two equal weights is equivalent to minimizing the pure Euclidean distance, that is, $w_Q = w_E = 1$, as equal weights do not affect the optimization problem in Eq. 1.

The results of the Split-Sample Test were benchmarked against the streamflow and evaporation outputs from the Integrated System for Rainfall-Runoff Modeling (SIMPA) model (Estrela and Quintas, 1996; Alvarez et al., 2005) for the study catchments. SIMPA is a well-established hydrologic model calibrated for the Spanish catchments and yearly updated which provides the foundation for water planning and management in the country (see https://www.miteco.gob.es/en/agua/temas/ evaluacion-de-los-recursos-hidricos/evaluacion-recursos-hidricos-regimen-natural.html for additional information). The SIMPA simulations are run at a monthly time step, and therefore the performances of VIC and SIMPA were compared for the Nash-Sutcliffe Efficiency of monthly streamflow ($NSE(Q_m)$) and $NSE(E_m)$.

Furthermore, the effect of considering the negative values in the streamflow series of reservoirs as null (see Section 2.2) was tested in an additional implementation of the Q-only calibration experiment for the 47 catchments with less than 5% of negative records and the 46 catchments with more than 5% of negative records (Fig. 2) after considering the negative values as gaps. This strategy made it possible to evaluate the extent to which the performance of VIC was affected by the data processing approach followed for the negative records.

Finally, a cross-validation test using multiple meteorological datasets was carried out to assess the generalizability of the calibrated parameters. Thus, the performance of the VIC model during the study period was further evaluated for the Q-only and Q-E calibration experiments using precipitation and temperature data gathered from a gridded dataset provided by the Spanish Meteorological Agency (AEMET, see https://www.aemet.es/en/serviciosclimaticos/cambio_climat/datos_diarios?w=2) and E-OBS (Cornes et al., 2018).





## 4 Results

### 4.1 RSA sensitivity analysis

The RSA sensitivity indices for $NSE(Q_d)$ and $NSE(E_m)$ are depicted in Fig. 4 and Fig. 5, respectively. $NSE(Q_d)$ sensitivities were mostly related to the five soil parameters and the two routing parameters (i.e. SR parameterization), with little or no

influence from the vegetation parameters and no clear spatial pattern for the RSA indices (Fig. 4). Among these parameters, the highest sensitivities corresponded to $d_2$, $rout_1$ and $rout_2$, although $b_i$, $D_S$, $W_S$ and $D_m$ were also influential to the streamflow metric according to several local estimates.

Contrarily to $NSE(Q_d)$, $NSE(E_m)$ scores were greatly influenced by the vegetation parameters (Fig. 5). In this case, the highest sensitivities corresponded to $rmin_f$ and $LAI_f$ and manifested a latitudinal gradient with minimum sensitivities occur-

ring for the northern catchments. This pattern is also noticeable, but to a lesser extent, for $depth_{1f}$, $rarc_f$, $albedo_f$, $rough_f$ and $RGL_f$. $d_2$ was revealed as the most important soil parameter to $NSE(E_m)$, and as expected from the VIC model implementation, the routing parameters had a null effect due to the routing scheme being exclusively applied to post-process the runoff simulations.

Fig. 6 shows the Spearman correlation coefficient ($r_S$) between the RSA sensitivity indices calculated for $NSE(Q_d)$ and

$NSE(E_m)$ and the physiographic and hydroclimatic characteristics in Fig. 3. The $NSE(Q_d)$ sensitivities for the soil parameters presented an opposite pattern (i.e., negative correlations) to mean annual precipitation, aridity index, NDVI and to a lesser extent slope. A similar behavior can be observed for various vegetation parameters such as $rmin_f$ and $LAI_f$, although they were not identified as important to $NSE(Q_d)$ (Fig. 4). Conversely, both routing parameters exhibited a matching pattern (i.e., positive correlations) to the previous four attributes but their magnitude was higher for $rout_1$. Concerning $NSE(E_m)$ sensitivities,

the soil parameters produced positive correlations with respect to those characteristics, whereas the vegetation parameters still reflected an opposite pattern. The correlations for mean temperature and saturated hydraulic conductivity ($K_S$) became noticeable for the $NSE(E_m)$ sensitivities and revealed an opposite pattern to the soil parameters and a matching pattern to most of the vegetation parameters.

The two most influential vegetation parameters to any of the two performance metrics under study were lastly selected

according to the values of the RSA index and were added to the SR parameterization during the calibration stage. Fig. 7 indicates that $LAI_f$ and $rmin_f$ were the two most influential parameters for the vast majority of the catchments with little influence from other vegetation parameters.

### 4.2 Split-Sample Test: calibration and evaluation

Fig. 8 shows the spatial distributions of $NSE(Q_d)$ and $NSE(E_m)$ corresponding to the Q-only (Fig. 8a,c) and Q-E (Fig. 8b,d)

calibration experiments for the calibration period. The relative gain/loss in model performance suggests that, while $NSE(Q_d)$ remained similar for both calibrations (Fig. 8a,b), $NSE(E_m)$ was highly improved after calibrating VIC against streamflow and evaporation data simultaneously (Fig. 8c,d).





This can also be appreciated in the CDFs of $\mathrm{NSE}(Q_\mathrm{d})$ and $\mathrm{NSE}(E_\mathrm{m})$ depicted in Fig. 9a,i. The median $\mathrm{NSE}(Q_\mathrm{d})$ was close to 0.6 for both experiments during the calibration period (Fig. 9a), although the streamflow performance was slightly deteriorated

for the Q-E calibration. On the other hand, the median $\mathrm{NSE}(E_\mathrm{m})$ for the Q-E calibration was 0.67 during the calibration period (Fig. 9i), while the median $\mathrm{NSE}(E_\mathrm{m})$ for the Q-only calibration did not exceed 0. As for the evaluation period, the slight to moderate loss in model performance for $\mathrm{NSE}(Q_\mathrm{d})$ and $\mathrm{NSE}(E_\mathrm{m})$ was indicative of an acceptable implementation and an adequate predictive capability.

The decomposition of $\mathrm{NSE}(Q_\mathrm{d})$ revealed similar $r_{Q_\mathrm{d}}$ estimates for both calibration experiments (Fig. 9b) and $\alpha_{Q_\mathrm{d}}$ values

generally below 1 (Fig. 9c). The $\beta_{Q_\mathrm{d}}$ distribution is approximately symmetric around the median for both calibrations but reflects a steeper CDF closer to 1 for the Q-only calibration (Fig. 9d). The $\mathrm{NSE}(E_\mathrm{m})$ improvement for the Q-E calibration is also evinced in its decomposition, $r_{E_\mathrm{m}}$ being the component subject to the greatest enhancement (Fig. 9j). $\alpha_{E_\mathrm{m}}$ and $\beta_{E_\mathrm{m}}$ estimates are comparable for both experiments with values slightly closer to 1 corresponding to the Q-E calibration (Fig. 9k,l), and point to a generalized overestimation of the variability and a slight underestimation of the bias, respectively.

The results of the Split-Sample Test were subsequently benchmarked against the performance of the SIMPA model for monthly streamflow (Fig. 9e-h) and monthly evaporation (Fig. 9i-l). Both calibration experiments outperformed SIMPA in terms of $\mathrm{NSE}(Q_\mathrm{m})$ and its decomposition, and although the poor performance of SIMPA for monthly evaporation was comparable to that of the Q-only calibration, the Q-E calibration produced much higher $\mathrm{NSE}(E_\mathrm{m})$ estimates.

Finally, the effect of handling the negative records in the streamflow series of 93 reservoirs (Fig. 2) was evaluated after

considering them as gaps and re-implementing the Q-only calibration experiment. Fig. 10a-d shows almost identical distributions for $\mathrm{NSE}(Q_\mathrm{d})$ and its decomposition for the 47 reservoirs with less than 5% of negative records, suggesting that the data processing strategy applied to the negative values had a minimum impact on the performance of VIC and thus corroborating the validity of considering them as null. This is also observable for the 46 reservoirs with more than 5% of negative records (Fig. 10e-h), although slight differences became apparent for the bias component (i.e., $\beta_{Q_\mathrm{d}}$) as the number of records modified

was greater.

### 4.3 Cross-validation test using multiple meteorological datasets

To cross-validate the results from the Split-Sample Test, the $Q/P$ bias was firstly calculated as the $Q/P$ ratio difference between the calibrated VIC and the observations using SPREAD/STEAD, AEMET and E-OBS data as the meteorological forcings of VIC for the complete study period (Fig. 11). $Q/P$ biases corresponding to the Q-only calibrated parameters were

broadly in the range $\pm 0.1$ for all the datasets (Fig. 11a-c), while the Q-E calibrated parameters produced increased deviations (Fig. 11d-f).

Results for SPREAD/STEAD and the Q-only calibration suggest that negative biases tend to be associated with higher $Q/P$ values and vice versa (see Fig. 11a and compare to Fig. 1b), whereas the $Q/P$ biases corresponding to the Q-E calibration display an opposite spatial distribution to that observed for the $(Q+E)/P$ values (see Fig. 11d and compare to Fig. 1c) and

exhibit a high negative correlation ($r = -0.91$). There is a predominance of negative $Q/P$ biases for both AEMET (Fig. 11b,e)





and E-OBS (Fig. 11c,f), even though these differences became exacerbated when forcing VIC with E-OBS and reached values below $-0.3$ in many of the northern catchments for the Q-E calibration.

The distributions of $NSE(Q_d)$, $NSE(E_m)$ and their decomposition for each meteorological dataset are depicted in Fig. 12. The performance attained for $NSE(Q_d)$ using SPREAD/STEAD was closely followed by AEMET, but reflected a moderate deterioration in the case of E-OBS (Fig. 12a). The daily streamflow dynamics were generally well captured with the three datasets (Fig. 12b), in particular with SPREAD/STEAD and AEMET, although an enhanced underestimation of the variability together with an underestimation of the bias became noticeable for AEMET and E-OBS (Fig. 12c,d).

The performance of VIC for $NSE(E_m)$, in turn, was similar for all the datasets and clearly demonstrated the effect of both calibration experiments (Fig. 12e). The monthly evaporation dynamics were well reproduced for the Q-E calibration in all cases (Fig. 12f), with estimates of the variability and the bias close to 1 (Fig. 12g,h).

## 5 Discussion

### 5.1 Parameter sensitivities

This study expands on previous investigations using VIC for the Duero River Basin in Yeste et al. (2020, 2021, 2023) by involving the main River Basin Districts in Spain. The large-sample approach followed in this work allowed for drawing more robust conclusions on model realism and the hydrologic functioning of a wide range catchments representing the hydroclimatic variability within the country. Parameter sensitivities were quantified according to the RSA sensitivity analysis method for the 189 study catchments and the various soil, routing and vegetation parameters indicated in Table 2. As in Yeste et al. (2023), sensitivities were calculated with respect to $NSE(Q_d)$ and $NSE(E_m)$, which were the performance metrics selected to evaluate the goodness-of-fit of VIC for streamflow and evaporation, respectively.

$d_2$ and the two routing parameters governing the gamma distribution function (i.e., $rout_1$ and $rout_2$) were identified as the most important parameters to $NSE(Q_d)$ (Fig. 4), highlighting the importance of applying a routing procedure to improve model performance for daily streamflow. The rest of the soil parameters were also identified as important to $NSE(Q_d)$ and yielded comparable sensitivities to those uncovered in previous studies (e.g., Gou et al., 2020; Lilhare et al., 2020; Melsen and Guse, 2019; Mendoza et al., 2015; Yeste et al., 2020, 2023). The influence of the vegetation parameters on $NSE(Q_d)$, however, was negligible, resembling the findings of Sepúlveda et al. (2022) for a large-sample application of VIC in Chile. The strong dependencies of the $NSE(Q_d)$ sensitivities for the soil and vegetation parameters on mean annual precipitation, aridity index and NDVI were manifested as either highly positive (i.e., a matching pattern) or highly negative (i.e., an opposite pattern) Spearman correlations (Fig. 6a), thus corroborating the interdependency between parameter sensitivities and climate variables found in Sepúlveda et al. (2022) for the Chilean catchments.

On the contrary, $NSE(E_m)$ was found to be most sensitive to the vegetation parameters, $LAI_f$ and $rmin_f$ being the most important vegetation parameters according to the RSA sensitivity indices (Fig. 5, 7). This is in line with the parameter sensitivities reported in Sepúlveda et al. (2022) and Yeste et al. (2023), suggesting that the VIC vegetation parameters have a significant potential to improve the representation of evaporative processes if included in model calibration. From among soil parameters,





$d_2$ was the most important parameter to NSE($E_\mathrm{m}$), which could be related to the water uptake by vegetation in the root zone as
it is directly affected by the thickness of the VIC soil layers.

## 5.2   Model performance during the Split-Sample Test and the cross-validation test

The large-sample application of the VIC model provided valuable insights into its performance for streamflow and evaporation
in the 189 study catchments. The capability of VIC to produce satisfactory estimates of NSE($Q_\mathrm{d}$) and NSE($E_\mathrm{m}$) simultaneously
was tested through a Split-Sample Test encompassing two calibration experiments based on the weighted Euclidean distance
definition for two objectives (Eq. 1), namely Q-only and Q-E calibration. While the Q-only calibration led to NSE($E_\mathrm{m}$) scores
below 0 for more than half of the catchments, the Q-E calibration substantially increased the performance for NSE($E_\mathrm{m}$)
and concomitantly produced NSE($Q_\mathrm{d}$) values closed to those corresponding to the Q-only calibration (Fig. 8, 9). Similar
conclusions were reached in Yeste et al. (2023) according to five single-objective calibration experiments carried out for one
test catchment located in the Guadalquivir River Basin.

The benchmark comparison of VIC against the performance of SIMPA for NSE($Q_\mathrm{m}$) and NSE($E_\mathrm{m}$) clearly indicated an
increased performance for both metrics when using VIC (Fig. 9). The monthly simulations from SIMPA stand as the greatest
modelling effort available for the Spanish domain and play a fundamental role in supporting water resource planning at the
national and river basin scales. Therefore, the implementation of VIC developed in this work constitutes an important leap
forward in comparison with the SIMPA simulations as VIC was run at a finer temporal resolution and improved the individual
and joint representation of streamflow and evaporation. Moreover, the performance of VIC for daily streamflow and monthly
evaporation was similar to that reflected in large-sample applications using VIC over the CONUS domain in Mizukami et al.
(2017) and Rakovec et al. (2019). The streamflow performance was also comparable to other modelling efforts involving the
Duero River Basin (Morán-Tejeda et al., 2014; Yeste et al., 2020, 2023), Tajo (Pellicer-Martínez and Martínez-Paz, 2018;
Pellicer-Martínez et al., 2021), Guadalquivir (Yeste et al., 2018), Segura (Pellicer-Martinez and Martínez-Paz, 2015; Pellicer-
Martínez et al., 2015) and Júcar (Marcos-Garcia et al., 2017; Suárez-Almiñana et al., 2020).

As described in Section 2.2, the presence of negative values in the streamflow series of 93 reservoirs (Fig. 2, 10) is likely
related to the indirect estimation of inflow data through a daily water balance of water storages and releases without considering
the evaporative fluxes from the reservoir. In this respect, the initial exploratory data analysis for the negative records represents
a call for action for future releases of the SAIH-ROEA dataset as it is not feasible to handle this issue on the basis of current
hydrologic information provided there. The effect of considering the negative values as null was evaluated during the Split-
Sample Test for the Q-only calibration experiment to quantify their relative significance in terms of model performance. The
distributions of NSE($Q_\mathrm{d}$) and its decomposition were virtually identical after considering the negative values both as null and
as gaps, suggesting that the simulated streamflow is also null or low during the warmest part of the hydrologic year.

Finally, the generalizability of the calibrated parameters was evaluated by means of a cross-validation test using meteoro-
logical information gathered from AEMET and E-OBS. As demonstrated in Yeste et al. (2023), the integration of streamflow
and evaporation data into model calibration is ultimately subject to the law of conservation of mass and the magnitude of the
imbalance stemming from merging three independent datasets of precipitation, streamflow and evaporation. This limitation





was thoroughly checked for the Q-only and Q-E calibration experiments and all the meteorological datasets, with results pointing to a higher $Q/P$ bias (Fig. 11) and a slight to moderate loss in model performance for the Q-E calibration (Fig. 12) as a
consequence of calibrating VIC against streamflow and evaporation data simultaneously. The potential of the calibrated parameters as well as the trade-off in model performance arising during the Q-E calibration experiment will be further explored in future implementations of VIC for the Spanish catchments to produce seamless distributed parameters maps and Spanish-wide simulations based on a fully gridded implementation.

## 6 Conclusions

In this work, a large-sample application of the VIC model was carried out for 189 headwater catchments belonging to the main River Basin Districts in Spain. The potential of combining streamflow and evaporation data into the hydrologic modelling exercise was explored for the sensitivity analysis stage, the calibration of the VIC parameters and the evaluation of its performance for the streamflow and evaporation simulations. The key findings of this study can be summarized as follows:

- A regional sensitivity analysis allowed for identifying the parameter sensitivities with respect to the selected metrics to
evaluate the performance of VIC against daily streamflow and monthly evaporation data for all the study catchments. The soil and routing parameters were revealed as the most important parameters to the streamflow performance, while the influence from the vegetation parameters was negligible. The performance of VIC for evaporation was mostly controlled by the soil parameters and two of the vegetation parameters.

- The calibration of the VIC model was performed by following two single-objective calibration experiments: a calibration
against daily streamflow data exclusively and a calibration against daily streamflow and monthly evaporation data simultaneously. The performance of VIC was assessed for two independent periods, suggesting that it is possible to achieve satisfactory adjustment to both hydrologic variables at the same time if their performance metrics are combined into a composite function based on a weighted Euclidean distance definition.

- A benchmark comparison was made between the performance of VIC and the monthly simulations from the SIMPA
model, the latter constituting the greatest modelling effort available to date for the Spanish domain. The VIC model led to an increased performance for both streamflow and evaporation compared to SIMPA, thus indicating a promising potential for a fully gridded implementation of VIC in the future to carry out Spanish-wide simulations.

- An additional evaluation of the performance of the VIC model was performed using meteorological observations from two independent gridded datasets in order to assess the generalizability of the calibrated parameters. The slight to mod-
erate loss in model performance at this stage was subject to the calibration experiment under study, with a greater imbalance and a trade-off in model performance becoming apparent for the calibration against streamflow and evaporation data simultaneously.



*Code and data availability.* Computer code for VIC (Liang et al., 1994, 1996) version 4.2.d can be downloaded from https://github.com/ UW-Hydro/VIC/tree/support/VIC.4.2.d. Scripts to perform the RSA sensitivity analysis are included in the SAFE Toolbox (Pianosi et al.,
2015). Precipitation and temperature data were collected from SPREAD (Serrano-Notivoli et al., 2017) and STEAD (Serrano-Notivoli et al., 2019). Streamflow time series were obtained from the SAIH-ROEA dataset (https://www.miteco.gob.es/en/cartografia-y-sig/ide/descargas/ agua/anuario-de-aforos.aspx). The required soil and vegetation parameters to run VIC were gathered from SoilGrids1km (Hengl et al., 2014) and EU-SoilHydroGrids ver1.0 (Tóth et al., 2017), and land uses were extracted from the UMD Global Land Cover Classification (Hansen et al., 2000). Data supplementing this study are available in the Zenodo repository https://doi.org/10.5281/zenodo.10670292 (Yeste et al.,
390 2024).

*Author contributions.* PY and MJEP developed the conceptualization and the methodology; PY carried out the analysis and wrote the original manuscript; MGVO helped process the data and analyse the results; AB provided valuable feedback on the methodology and the results; SRGF, YCD and MJEP contributed to the editing of the manuscript, supervised the project and acquired the funding.

*Competing interests.* The authors declare that they have no conflict of interest.

*Acknowledgements.* The first author acknowledges the Alexander von Humboldt Foundation for a Humboldt Research Fellowship for postdoctoral researchers and the Ministry of Education, Culture and Sport of Spain for a FPU grant (reference FPU17/02098) during his doctoral training. This work was supported by the project P20_00035, funded by the FEDER/Junta de Andalucía - Consejería de Transformación Económica, Industria, Conocimiento y Universidades, the project CGL2017-89836-R funded by the Spanish Ministry of Economy and Competitiveness, with additional support from the European Community Funds (FEDER), the project PID2021-126401OB-I00, funded by
MCIN/AEI/10.13039/501100011033/FEDER Una manera de hacer Europa and the project LifeWatch-2019-10-UGR-01 funded by FEDER/Ministerio de Ciencia e Innovación. All the simulations were conducted in the ALHAMBRA cluster (https://supercomputacion.ugr.es/) of the University of Granada. ChatGPT 3.5 was used to suggest rewording of the VIC model description in the Methods section and avoid redundancy with respect to previous publications from the first author using VIC.



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



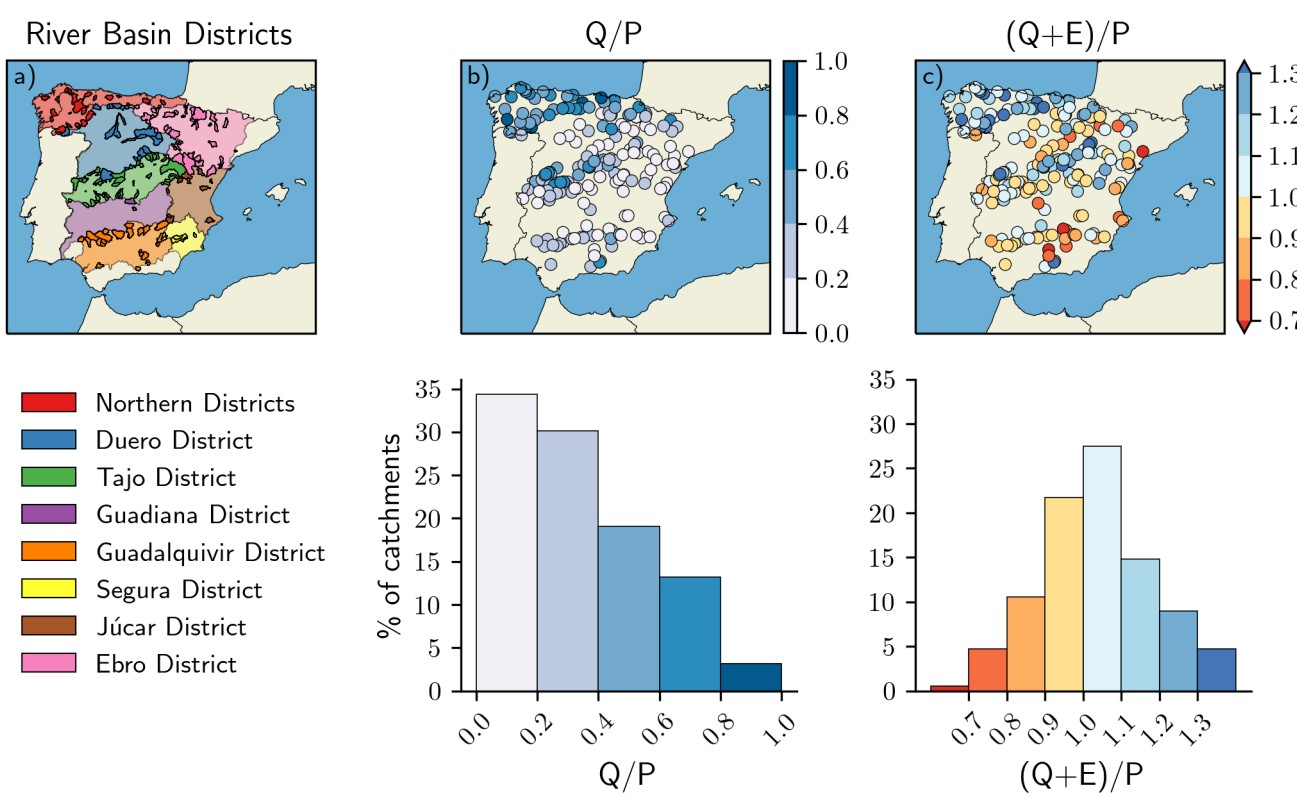

**Figure 1.** (a) Topographic boundaries of the 189 headwater catchments and the main River Basin Districts in Spain. (b) Spatial distribution of the runoff ratio ($Q/P$). (c) Spatial distribution of the sum of the runoff and the evaporation ratios to precipitation ($(Q+E)/P$). Values in (b) and (c) were calculated for the study period and the hydroclimatic datasets considered in this work.



**Figure 2.** Exploratory data analysis of the negative values in the daily time series of streamflow gathered from the SAIH-ROEA dataset for the 189 studied catchments during the study period. (a) Scatter plot representation of the percentage ratio of negative to positive values (y-axis) against the percentage ratio of the number of negative values to the number of records (x-axis) for each catchment. "*R*" denotes "Reservoir" and "*GS*" denotes "Gauging Station". (b) Monthly distribution of the percentage ratio of the number of negative values per month to the total number of negative values. Blue line corresponds to the average hydrologic year of streamflow expressed as the monthly mean percentage of annual streamflow calculated over all the catchments.





**Figure 3.** Spatial distribution of the physiographic and hydroclimatic characteristics analysed in this work as potential drivers of parameter sensitivities. These attributes are defined in Table 3.



**Figure 4.** Spatial distribution of the RSA sensitivity indices calculated for NSE($Q_d$).







**Figure 5.** Spatial distribution of the RSA sensitivity indices calculated for NSE($E_{\mathrm{m}}$).

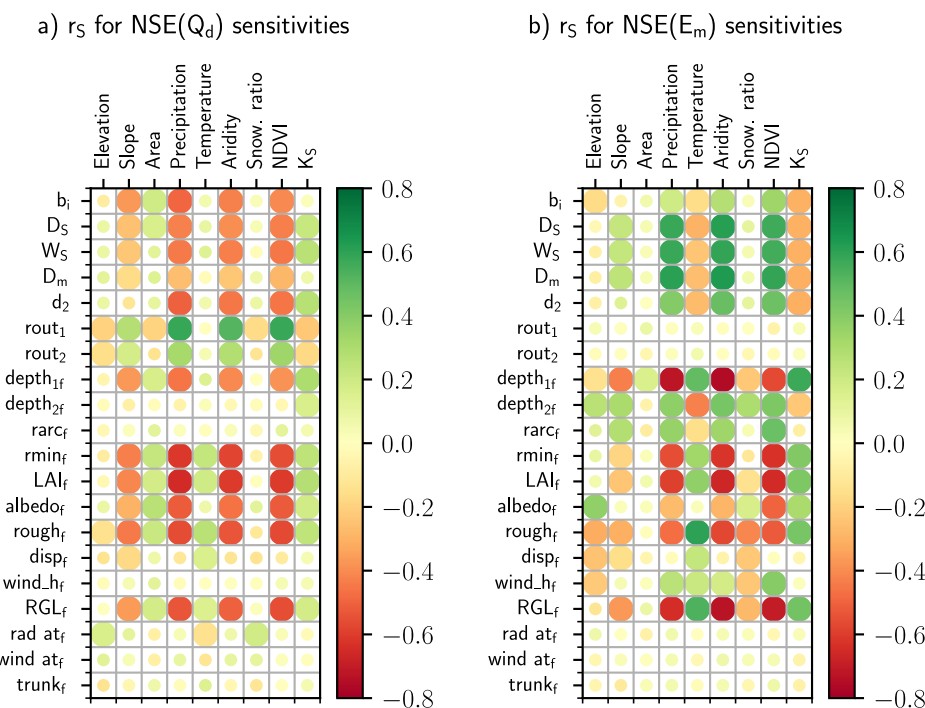

**Figure 6.** Spearman correlation coefficient ($r_S$) between the RSA sensitivity indices calculated for (a) NSE($Q_d$) and (b) NSE($E_m$) and the physiographic and hydroclimatic characteristics defined in Table 3 and depicted in Fig. 3. Full-size circles indicate statistically significant $r_S$ estimates at the 95% confidence level.





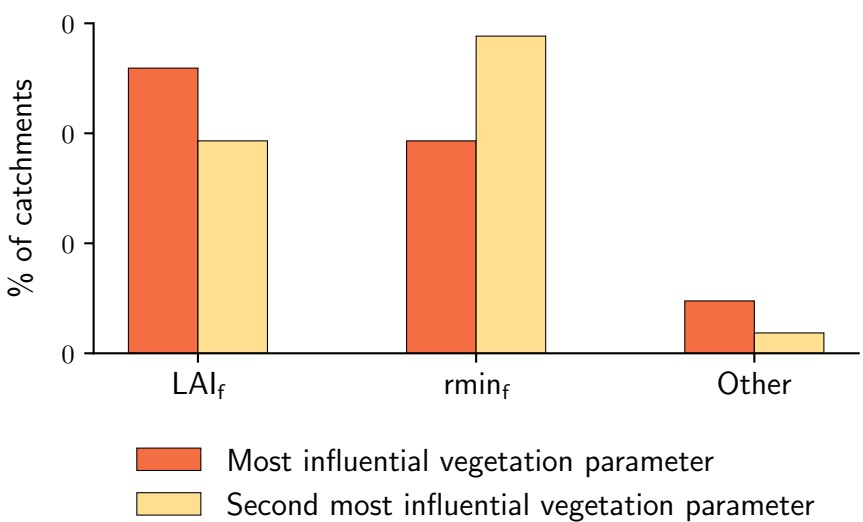

**Figure 7.** Frequency of the (first) most influential and second most influential vegetation parameters according to the RSA sensitivity indices calculated for $\mathrm{NSE}(Q_\mathrm{d})$ and $\mathrm{NSE}(E_\mathrm{m})$.



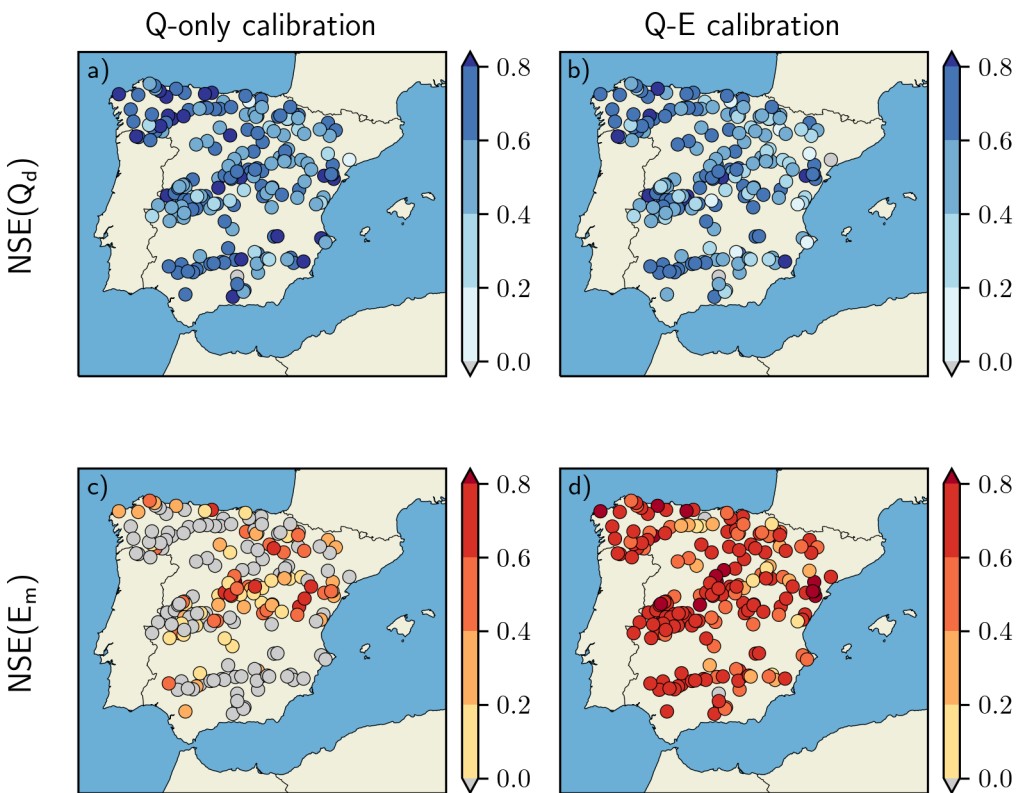

**Figure 8.** Spatial distribution of $NSE(Q_d)$ and $NSE(E_m)$ for (a,c) the Q-only calibration and (b,d) the Q-E calibration during the calibration period.

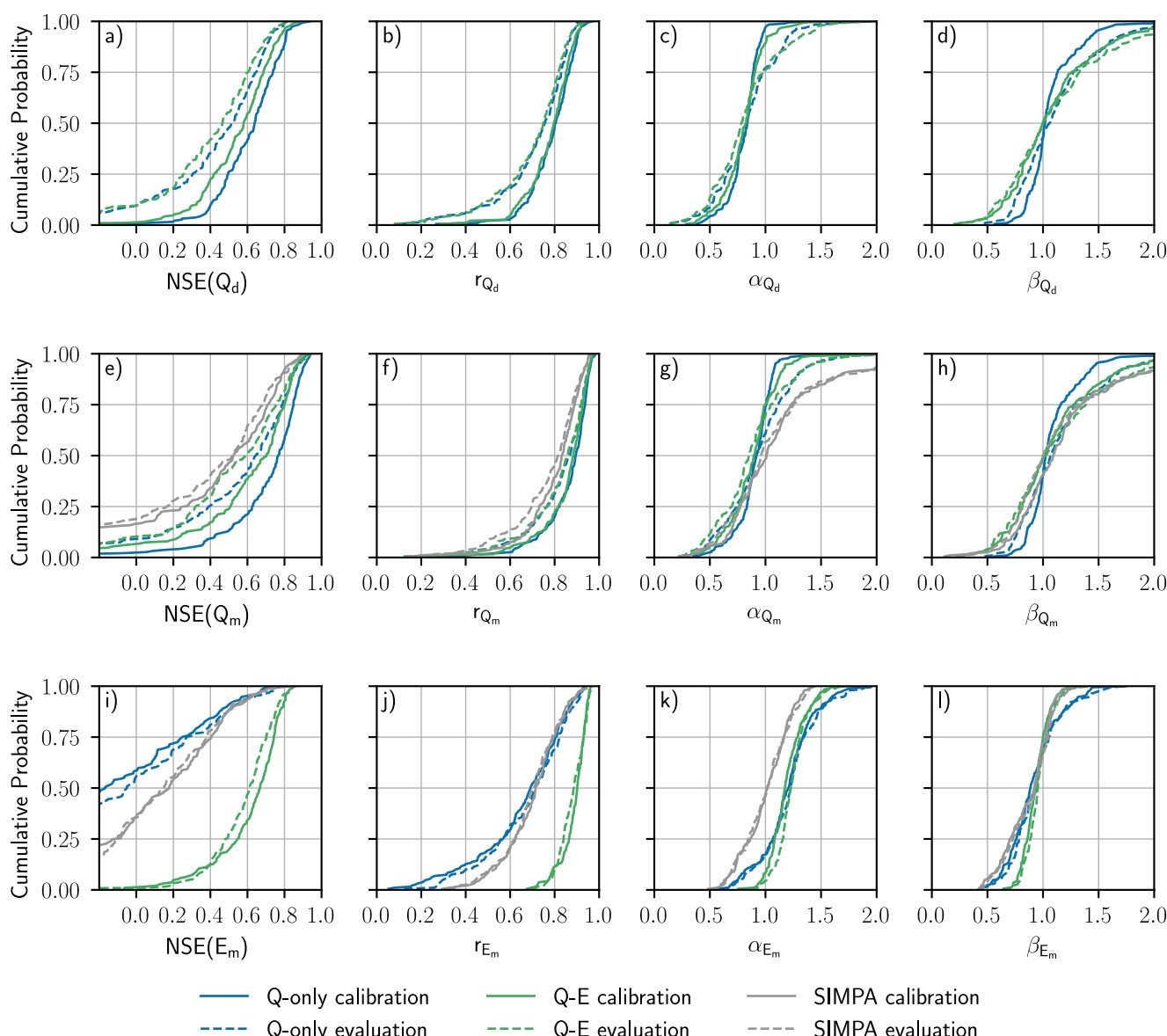

**Figure 9.** CDFs of NSE and its decomposition calculated for (a-d) daily streamflow, (e-h) monthly streamflow and (i-l) monthly evaporation. Blue lines and green lines indicate the performance of VIC for the Q-only and Q-E calibration experiments, respectively, while grey lines correspond to the performance of SIMPA (note that the SIMPA simulations are only available at a monthly time step). Results for the calibration (evaluation) period are represented with solid (dashed) lines.

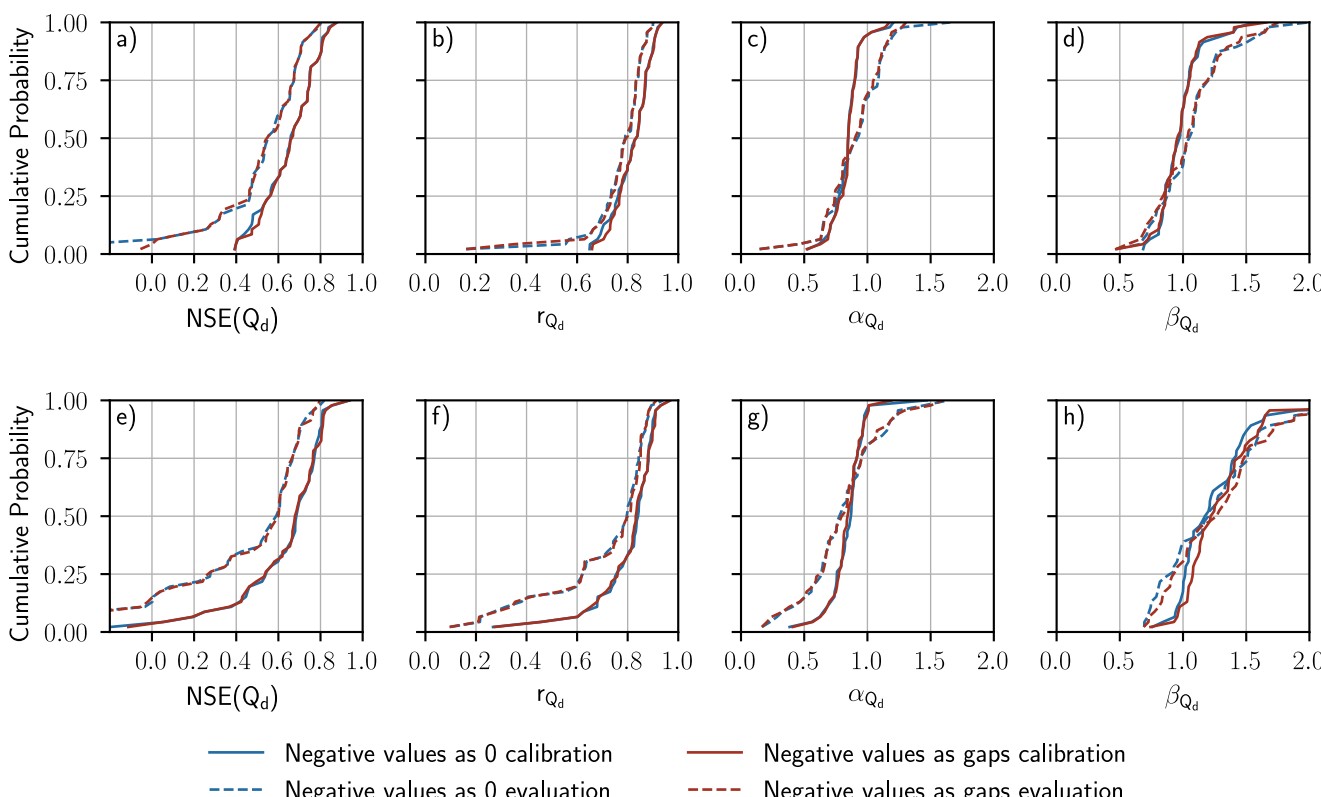

**Figure 10.** CDFs of NSE($Q_d$) and its decomposition for the Q-only calibration experiment in reservoirs presenting negative records of daily streamflow: (a-d) 47 reservoirs with less than 5% of negative records and (e-h) 46 reservoirs with more than 5% of negative records (see also Fig. 2). Dark blue lines and light blue lines indicate the performance of VIC after considering the negative values as 0 and as gaps, respectively. Results for the calibration (evaluation) period are represented with solid (dashed) lines.

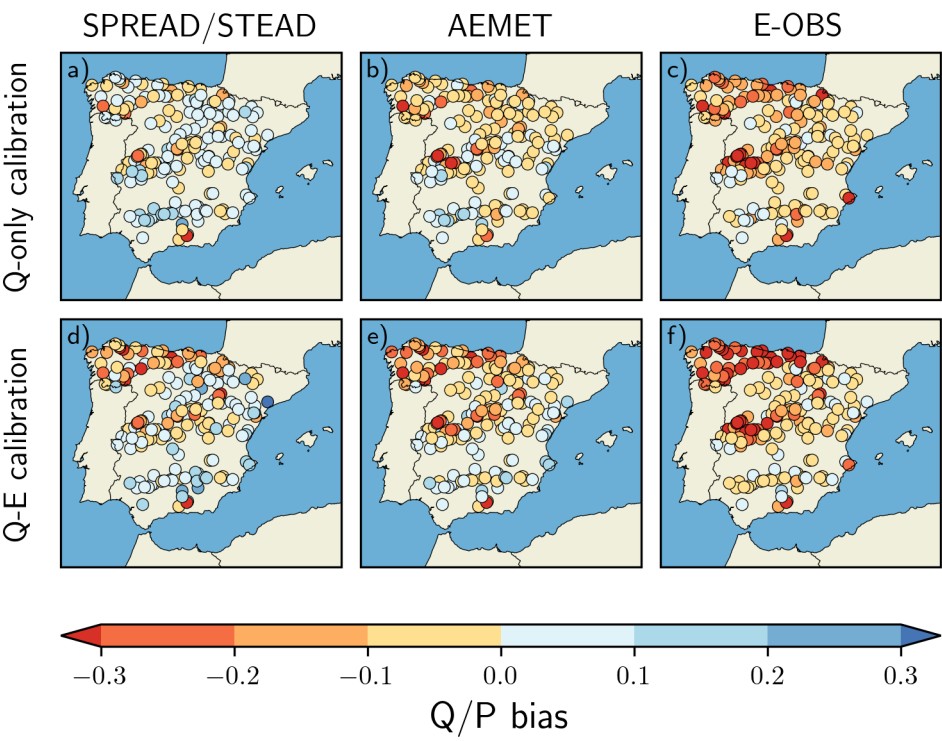

**Figure 11.** Spatial distribution of the $Q/P$ bias calculated as the $Q/P$ ratio difference between the calibrated model and the observations (see Fig. 1b) using meteorological data from SPREAD/STEAD, AEMET and E-OBS to force VIC for the complete study period. (a to c) $Q/P$ bias corresponding to the Q-only calibrated parameters. (d to f) $Q/P$ bias corresponding to the Q-E calibrated parameters.

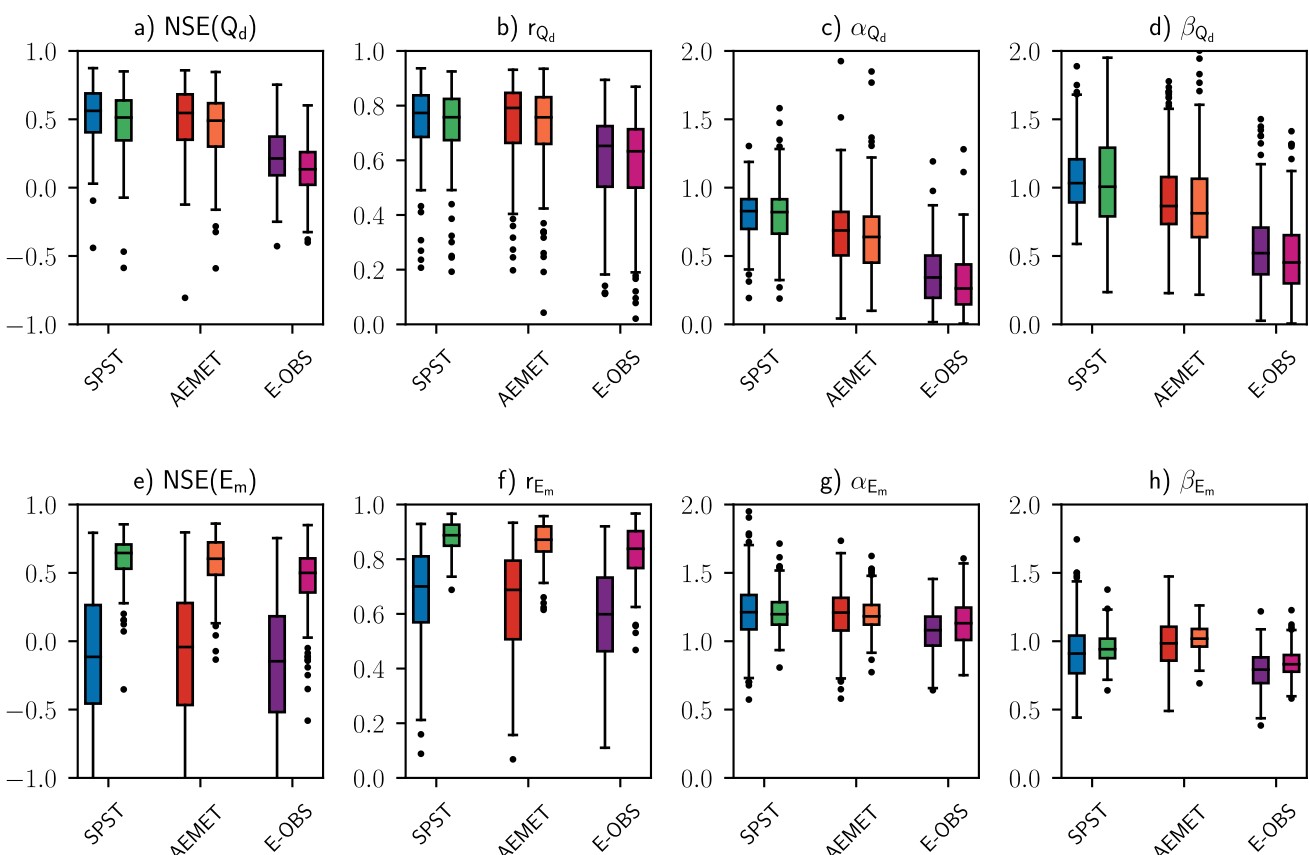

**Figure 12.** Distributions of NSE and its decomposition calculated for (a-d) daily streamflow and (e-h) monthly evaporation using the SPREAD/STEAD (SPST), AEMET and E-OBS datasets for the complete study period. Blue, red and purple boxplots (i.e., boxplots to the left in each pair group) correspond to the Q-only calibration experiment. Green, orange and pink boxplots (i.e., boxplots to the right in each pair group) correspond to the Q-E calibration experiment.





**Table 1.** Number of headwater reservoirs and gauging stations per River Basin District included in this study.

| River Basin District | Reservoirs | Gauging stations | Total |
|---|---|---|---|
| Northern Districts | 13 | 15 | 28 |
| Duero District | 11 | 12 | 23 |
| Tajo District | 14 | 35 | 49 |
| Guadiana District | 2 | 0 | 2 |
| Guadalquivir District | 27 | 0 | 27 |
| Segura District | 6 | 0 | 6 |
| Júcar District | 8 | 5 | 13 |
| Ebro District | 13 | 28 | 41 |
| **Total** | **94** | **95** | **189** |



**Table 2.** Parameters included in the RSA sensitivity analysis. As in Yeste et al. (2023), subscript "f" corresponds to the VIC vegetation parameters that were modified using dimensionless multiplication factors.

| Parameter | Units | Min value | Max value | Description |
|---|---|---|---|---|
| $b_i$ | - | $10^{-5}$ | 0.4 | Variable infiltration shape parameter |
| $D_S$ | - | $10^{-9}$ | 1 | Fraction of $D_m$ where non-linear baseflow begins |
| $W_S$ | - | $10^{-9}$ | 1 | Fraction of the porosity of the bottom soil layer where non-linear baseflow begins |
| $D_m$ | mm/day | $10^{-9}$ | 30 | Maximum baseflow |
| $d_2$ | m | 0.1 | 0.9 | Thickness of soil layer 2 |
| $rout_1$ | - | $10^{-9}$ | 10 | Shape parameter of gamma function |
| $rout_2$ | days | $10^{-9}$ | 2 | Scale parameter of gamma function |
| $depth_{1f}$ | - | 0.5 | 1.5 | Thickness of root zone for layer 1 |
| $depth_{2f}$ | - | 0.5 | 1.5 | Thickness of root zone for layer 2 |
| $rarc_f$ | - | 0.5 | 1.5 | Architectural resistance |
| $rmin_f$ | - | 0.5 | 1.5 | Minimum stomatal resistance |
| $LAI_f$ | - | 0.5 | 1.5 | Leaf-area index |
| $albedo_f$ | - | 0.5 | 1.5 | Albedo |
| $rough_f$ | - | 0.5 | 1.5 | Vegetation roughness |
| $disp_f$ | - | 0.5 | 1.5 | Vegetation displacement |
| $wind\_h_f$ | - | 0.5 | 1.5 | Height of wind speed measures |
| $RGL_f$ | - | 0.5 | 1.5 | Minimum incoming shortwave radiation for transpiration |
| $rad\_atten_f$ | - | 0.5 | 1.5 | Radiation attenuation |
| $wind\_atten_f$ | - | 0.5 | 1.5 | Wind speed attenuation through overstory |
| $trunk\_ratio_f$ | - | 0.5 | 1.5 | Ratio of total tree height that is trunk |





**Table 3.** Definition of the physiographic and hydroclimatic characteristics analysed in this work as potential drivers of parameter sensitivities.

| Characteristic | Type | Definition | Source |
|---|---|---|---|
| Elevation | Physiographic | Mean catchment elevation (m.a.s.l.) | Digital Elevation Model (DEM) EU-DEM 30m (now at https://spacedata.copernicus.eu/collections/copernicus-digital-elevation-model) |
| Slope | Physiographic | Mean catchment slope (m/km) | Derived from DEM |
| Area | Physiographic | Catchment area ($km^2$) | Derived from DEM |
| Precipitation | Hydroclimatic | Mean annual precipitation (mm/yr) | SPREAD (Serrano-Notivoli et al., 2017) |
| Temperature | Hydroclimatic | Mean temperature (ºC) | STEAD (Serrano-Notivoli et al., 2019) |
| Aridity | Hydroclimatic | Aridity index, calculated as the ratio of precipitation to potential evaporation (-) | Potential evaporation calculated internally in VIC; precipitation from SPREAD (Serrano-Notivoli et al., 2017) |
| Snowfall ratio | Hydroclimatic | Snowfall ratio to precipitation during winter months (i.e. December, January and February) (-) | Snowfall calculated internally in VIC; precipitation from SPREAD (Serrano-Notivoli et al., 2017) |
| NDVI | Physiographic | Normalized Difference Vegetation Index (-) | Copernicus Global Land Service (https://land.copernicus.eu/global/products/NDVI) |
| $K_S$ | Physiographic | Saturated hydraulic conductivity (mm/d) | EU-SoilHydroGrids (Tóth et al., 2017) |