# Peer review of "A large-sample modelling approach towards integrating streamflow and evaporation data for the Spanish catchments"

_Hydrology and Earth System Sciences, 2024_

## Author Comment (AC1)

**Responses to Anonymous Referee #1**

This study investigates the capacity of the VIC model to integrate streamflow and evaporation data in a large-sample application, based on simulations for 189 headwater catchments in Spain. Utilizing multiple datasets to improve model performance is an important aspect of hydrological modeling research, making this paper potential for publication in HESS. However, some important issues are not discussed adequately, and there is room for improvement. Consequently, I recommend a major revision before publication.

We thank the reviewer for his/her constructive feedback and we are convinced it will contribute to improve the manuscript. We have indicated in our responses those references that were not included in the initial version of the manuscript.

My major suggestions are as follows:

Given the general imbalances in Q, E, and P, I question whether it is reasonable to use the evaporation dataset to evaluate the model directly. This brings another question: what is the most important signature provided by the evaporation data? Is it the evaporation amount or the temporal variation of evaporation? If the key information provided by the evaporation data is the amount, we can expect that the E simulation obtained by Q-only calibration will be good in the catchments with Q+E/P close to 1, and the NSE(Q) obtained by Q-E calibration will be lower than that from Q-only calibration (perhaps the authors can test whether the results indeed show this characteristic). Otherwise, if the key information provided by the evaporation data is the variation, conducting a bias correction on E data based on water balance before calibration makes more sense.

We thank the reviewer for pointing out this important issue regarding the imbalances in Q, E and P. In our understanding, the key information provided by the evaporation dataset is the amount of evaporation. This does not mean that the temporal dynamics are not important, but they are evaluated at the monthly time scale and thus are less relevant than in the case of streamflow, which is evaluated at daily scale. The amount of evaporation together with streamflow and precipitation allow for identifying gaining and losing catchments as they constitute an indirect measure of the intercatchment groundwater flow (Liu et al., 2020). Gaining and losing catchments are therefore characterized by an unclosed water balance, which can potentially lead to an unrealistic representation of the partitioning of precipitation into streamflow and evaporation in case of significant imbalances when using a (closed) water balance hydrologic model.

This issue was addressed in our original submission by performing two calibration experiments, namely Q-only and Q-E calibration, in order to assess the relative gain/loss in model performance in terms of NSE(Q) and NSE(E). Results showed that NSE(Q) remained similar for both calibrations, indicating that the imbalances in Q, E and P did not deteriorate model performance for streamflow in a substantial manner. However, we agree that there is room for improvement and that this issue is worthy of further examination. As suggested by the reviewer, we have evaluated the performance for Q and E during both calibration experiments considering the (Q+E)/P ratio as a signature of how far/close is the

water balance from being closed from a data perspective. We have performed the analysis for NSE as well as for its decomposition into r, α, and β components for the 189 study catchments (please, see also our response in relation to the decomposition of NSE into r, α, and β). The results are depicted in the following figure:

[Figure]

Panels a-d) correspond to the Q-only calibration and panel e-h) correspond to the Q-E calibration, and the different values have been calculated for the complete study period. As rightly pointed out by the reviewer, the NSE(E) estimate obtained with the Q-only calibration is better for the catchments with (Q+E)/P close to 1 (panel a), and the NSE(Q) produced by the Q-E calibration is lower than that from the Q-only calibration (panel e). As suggested in Yeste et al. (2023), the β component (i.e., the bias component) is of capital importance from a water balance perspective as it is sensitive to the imbalances of the Q, E and P data when Q and E are integrated into model calibration. As shown in panels d) and h), β is closer to 1 for both Q and E for catchments with (Q+E)/P close to 1, with a wider distribution of $\beta_Q$ for the Q-E calibration due to the imbalances of the Q, E and P data. These imbalances, however, do not have a marked effect for the dynamics (i.e., r) and the variability (i.e., α), as shown in panels b) and f), and c) and g), respectively. Hence, although NSE(Q) values are slightly lower for the Q-E calibration, they remain similar for both calibration experiments, as indicated before.

This figure will be introduced in Section 4.2 and will be discussed in Section 5.2 in the revised version of the manuscript.

Liu, Y., Wagener, T., Beck, H. E., & Hartmann, A. (2020). What is the hydrologically effective area of a catchment? Environmental Research Letters, 15(10). https://doi.org/10.1088/1748-9326/aba7e5

An important role of adopting multiple datasets is to reduce equifinality, i.e., to reject some parameters that perform well in the simulation of only one objective. However, the authors didn't address this issue, stopping at presenting good simulations for streamflow and evaporation. I encourage the authors to discuss the value of the evaporation dataset in reducing equifinality. A potential way to address this issue is to analyze the sensitivity of the integrated objective function to the parameters and compare it with the sensitivity of NSE(Q).

This is ideally done following a Pareto optimization approach for both NSE(Q) and NSE(E) as in Yeste et al. (2023). Pareto optimization is known for reducing equafinality as the model is optimized for two or more objective functions simultaneously, resulting in a lower number of behavioural parameter sets (Efstratiadis and Koutsoyiannis, 2010). Although we have not implemented a full Pareto optimization, it is still possible to address this issue only considering the two calibration experiments in this work as they represent two important solutions belonging to the Pareto front: the corner corresponding to the maximum NSE(Q) performance and the compromise solution for NSE(Q) and NSE(E) considering a weighted Euclidean distance with equal weights for both.

We value the suggestion from the reviewer to compare the sensitivity indices of NSE(Q) and the integrated objective function. However, the sensitivities indices are based on the Monte Carlo experiment comprising 10000 Latin Hybercube samples, and therefore they are representative of the complete parameter and objective space. As equafinality is concerned with behavioural parameter combinations, we propose that such comparison should be carried out only for the region where the best candidate solutions are located.

Following this reasoning, we have calculated for every catchment the mean absolute deviation of NSE(Q) from the maximum NSE(Q) considering the 1%, 2% and 5% best performing simulations from the Monte Carlo experiment and according to two criteria: 1) NSE(Q) itself and 2) the Euclidean distance for NSE(Q) and NSE(E). Results are shown in the following figure:

[Figure]

The boxplot above shows two effects: 1) as the percentage of best performers considered increases, the deviations from the maximum NSE(Q) become higher; 2) the deviations are

more pronounced when the best performing criterion is based on the Euclidean distance for NSE(Q) and NSE(E). The first effect is a straightforward consequence of considering an increasing number of simulations to calculate the mean absolute deviation from the maximum NSE(Q). The second effect is an indicator of less equifinality as there are fewer parameter combinations yielding a performance close to the maximum NSE(Q), which highlights the value of using multiple datasets in reducing equifinality.

This figure will be introduced in Section 4.1 and will be discussed in Section 5.1 in the revised version of the manuscript.

Efstratiadis, A., & Koutsoyiannis, D. (2010). One decade of multi-objective calibration approaches in hydrological modelling: a review. Hydrological Sciences Journal, 55(1), 58–78. https://doi.org/10.1080/02626660903526292

Other minor and moderate issues:

- Data

There are negative records at some stations, which can be attributed to the reservoir. So a question is whether the influence of the reservoir on streamflow is significant and whether the reservoir is simulated in the model. To my knowledge, some rivers significantly influenced by reservoirs have an extremely even interannual streamflow distribution, which is impossible to reproduce if the reservoir is not considered in the model.

The streamflow dataset is not affected by river regulation as the study catchments selected in this work are located in headwater areas and there are no dams within them. Some of the catchments were delineated considering the reservoirs as outlets, and these are the stations that the reviewer is referring to. As described in Section 2.1, the streamflow time series for reservoirs were indirectly estimated as a water balance between water storages and releases. The regulated component corresponds to the water release, which is subtracted from the water storage to calculate the input flow to the reservoir. The latter is then equated with the streamflow of the catchment, and therefore there is no influence of the reservoir on streamflow as the input flow to the reservoir is not regulated.

- Methods

The model performance was evaluated using NSE and its decomposition into γ, α, and β. However, to my understanding, γ, α, and β are the decomposition of KGE rather than NSE. NSE actually only quantifies the bias characteristic. I suggest the authors modify this expression. Additionally, please add the equations for these three metrics.

Both KGE and NSE can be decomposed into r (correlation coefficient), α (variability term), and β (bias term), and therefore both constitute integrative metrics of model performance. Thus, NSE not only quantifies the bias characteristic (i.e., β), but also the dynamics (i.e., r) and the variability (i.e., α). The expression for NSE as a function of r, α, and β is given by (e.g., Knoben et al., 2019):

$$NSE = 2\,\alpha r - \alpha^2 - \frac{(\beta - 1)^2}{CV_{obs}^2}$$

where $CV_{obs}$ is the coefficient of variation of the observations. The equation of NSE as a function of r, α, and β and the equations for α, and β will be indicated in the revised version of the manuscript. The correlation coefficient does not need to be defined given its widespread use.

Knoben, W. J. M., Freer, J. E., & Woods, R. A. (2019). Technical note: Inherent benchmark or not? Comparing Nash–Sutcliffe and Kling–Gupta efficiency scores. Hydrology and Earth System Sciences, 23(10), 4323–4331. https://doi.org/10.5194/hess-23-4323-2019

I am a little confused about the SST test. If I understand correctly, this seems to be the common practice in model calibration, i.e., to divide calibration and evaluation periods, with a warm-up period in each. Please correct me if I am wrong, but if this is indeed the common practice, I suggest not referring to it using a special term.

The pioneering work of Klemeš (1986) defines two splitting strategies: the Split-Sample Test (SST) and the Differential Split-Sample Test (DSST). Both approaches have been extensively used in hydrology, and the terms SST and DSST are specifically referred to in a large number of hydrologic studies (e.g., Fowler et al., 2018, 2021; Gharari et al., 2013; Melsen et al., 2019; Rakovec et al., 2019) as it is a direct way to indicate which calibration approach has been applied. Therefore, we consider that it is important to use the term SST as it can help the hydrologic community to clearly identify the practice followed in our study. In alignment with all the previous studies in which this terminology is specifically used, we have decided to keep our reference to the SST test.

Fowler, K., Coxon, G., Freer, J., Peel, M., Wagener, T., Western, A., Woods, R., & Zhang, L. (2018). Simulating Runoff Under Changing Climatic Conditions: A Framework for Model Improvement. Water Resources Research, 54(12), 9812–9832. https://doi.org/10.1029/2018WR023989

Fowler, K. J. A., Coxon, G., Freer, J. E., Knoben, W. J. M., Peel, M. C., Wagener, T., Western, A. W., Woods, R. A., & Zhang, L. (2021). Towards more realistic runoff projections by removing limits on simulated soil moisture deficit. Journal of Hydrology, 600(December 2020), 126505. https://doi.org/10.1016/j.jhydrol.2021.126505

Gharari, S., Hrachowitz, M., Fenicia, F., & Savenije, H. H. G. (2013). An approach to identify time consistent model parameters: sub-period calibration. Hydrology and Earth System Sciences, 17(1), 149–161. https://doi.org/10.5194/hess-17-149-2013

Melsen, L. A., Teuling, A. J., Torfs, P. J. J. F., Zappa, M., Mizukami, N., Mendoza, P. A., Clark, M. P., & Uijlenhoet, R. (2019). Subjective modeling decisions can significantly impact the simulation of flood and drought events. Journal of Hydrology, 568(September 2017), 1093–1104. https://doi.org/10.1016/j.jhydrol.2018.11.046

- Sensitivity Analysis

The authors calculate the sensitivity of model performance to parameters and analyze the correlation between sensitivity and physiographic hydroclimatic characteristics. I think it would be interesting to discuss the mechanism behind this correlation, e.g., why NSE(Q) is more sensitive to rout1 in catchments with larger precipitation. This can also provide guidance on selecting sensitivity parameters in regions with different conditions. I

encourage the authors to delve deeper into this analysis or discussion. Currently, this is only discussed by comparing with other studies, without addressing the underlying reasons.

We thank the reviewer for his/her suggestion. We think that investigating the mechanisms behind the correlations between the parameter sensitivities and the physiographic and hydroclimatic characteristics will positively contribute to improve the manuscript. The following discussion revolves around two axes: 1) NSE(Q) sensitivities, 2) NSE(E) sensitivities.

- NSE(Q) sensitivities: as stated in Section 4.1, the highest NSE(Q) sensitivities correspond to the soil parameters ($b_i$, $D_s$, $W_s$, $D_m$ and $d_2$) and the routing parameters ($rout_1$ and $rout_2$) (Fig. 4). The soil parameters presented an opposite pattern (i.e., negative correlations) to mean annual precipitation, aridity index, NDVI and to a lesser extent slope (Fig. 6). These parameters control the runoff generation process in VIC, and the negative correlations indicate that they are more important for catchments characterized by a more arid climate. As the precipitation volume to be transformed into runoff is lower for such catchments, the role of the five soil parameters becomes critical in modulating the runoff generation, whereas their effect is less relevant for catchments belonging to a more humid climate given the higher water availability. The generated runoff volume is subsequently routed to the catchment outlet according to a gamma-based unit hydrograph in a post-processing phase. The two routing parameters control the delay between runoff generation and catchment discharge (i.e., streamflow) and exhibited a matching pattern (i.e., positive correlations) to the previous four attributes (Fig. 6), suggesting that both parameters are important for the humid catchments as a consequence of the higher runoff volumes to be routed.

- NSE(E) sensitivities: NSE(E) was greatly influenced by the vegetation parameters, in particular by $rmin_f$ and $LAI_f$ (Fig. 5). Among the soil parameters, $d_2$ was revealed as the most important soil parameter to NSE(E), and as expected, the routing parameters showed a null effect. The high NSE(E) sensitivities for $rmin_f$ and $LAI_f$ reflected negative correlations to mean annual precipitation, aridity index and NDVI (Fig. 6), denoting a greater impact for arid catchments that is likely associated to the limiting effect on the evaporative processes entailed by a lower vegetation density. As for the soil parameters, the high NSE(E) sensitivities for $d_2$ could be related to the water uptake by vegetation in the root zone as it is directly affected by the thickness of the VIC soil layers. The positive correlations associated to the five soil parameters with respect to the previous characteristics are likely connected to the implementation of the closed water balance equation VIC and manifest an opposite behaviour to that observed NSE(Q). This effect was also appreciated in Yeste et al. (2020) for the sensitivities of the VIC soil parameters.

The previous discussion will be integrated in Section 5.1 in the revised version of the manuscript.

The Y-axis of Figure 7 is incorrect.

We thank the reviewer for pointing out this mistake. It will be modified in the revised version of the manuscript.

---

## Author Comment (AC2)

**Responses to Ilhan Özgen-Xian**

Summary. This article investigates how using evapotranspiration in addition to streamflow data during the calibration process affects the model performance across 189 Spanish catchments. The Variable Infiltration Capacity (VIC) model, a semi-distributed hydrological model, is used for modelling. For the calibration of evapotranspiration fluxes, leaf area index and minimal stomatal conductivity were found to be the most sensitive parameters. Apart from the vegetation parameterisation, soil parameters were found to have a large effect on model results. The VIC model is run for 20 years (with a spin-up period of 10 years) using different targets for the calibration. Results are analysed and discussed with a focus on model sensitivity to calibration target.

Assessment. The subject of this article is interesting. Indeed, large sample and large scale hydrological modelling is becoming more feasible, due to improvements in computing technology. In this sense, the topic is timely and of interest. The manuscript is well written and easy to follow. I think the modelling work is substantial and the work is suitable to be published in HESS. I have some questions that I would like the authors to address. These are listed below. I recommend minor revision before publication.

We thank the reviewer for his positive feedback and interesting questions as they will contribute to improve the manuscript. We have indicated in our responses those references that were not included in the initial version of the manuscript.

Comment on large sample modelling in this work.

The use of large samples in hydrology is very interesting. Especially when combined with a comparative analysis, the large sample size can help us to discover relations among processes and generate hydrological insights. While this work made use of data from a large number of catchments, the discussion of the results were focused on quite technical issues of model structure and calibration. I do not think this is bad, in fact, these are important topics. But I wonder whether the conclusion in the abstract, i.e. "This investigation will help gain a better understanding of the hydrology of the Spanish catchments and will help prepare the ground for a fully gridded implementation of the VIC model in Spain." still holds. Perhaps in the conclusions, the authors could provide some insights of the hydrology of Spanish catchments they gained in this study. Or perhaps this manuscript focuses on model sensitivity, in this case, this should be better reflected in the abstract.

We thank the reviewer for his positive feedback regarding the use of large samples in hydrology. Indeed, the importance of large-sample hydrology lies on the analysis of multiple catchments to draw general conclusions about the hydrologic functioning of the domain under investigation. We expect that the results from this work will be the basis for a future implementation of VIC for the Spanish catchments and will contribute to produce seamless distributed parameters maps and Spanish-wide simulations based on a fully gridded implementation. However, this is a future development and not a current result from this work. Therefore, we apologize for concluding in the abstract in such a way. As the

manuscript focuses on model sensitivity and model performance during calibration, this will be better reflected in the abstract in the revised version of the manuscript.

Questions.

1. Here is my understanding of the modelling work, please correct me if I am wrong: The VIC model is a semi-distributed hydrological model in the sense that no horizontal fluxes are computed between individual grid cells. VIC is set up for entire Spain, thus, all catchments included in this studies data set are represented in the model. Model calibration is done by adjusting model parameters in each grid cell individually.

The description of the VIC model is correct. However, VIC was not implemented for entire Spain, but rather for the individual catchments with a spatial resolution of 5 km for the grid cells overlapping the catchment area. As for the calibration, the calibrated parameters were considered as spatially constant for all the cells affecting the catchment. The rest of the soil and vegetation parameters not modified during calibration were derived for each cell individually. This approach was also followed in previous implementations of VIC in Yeste et al. (2020, 2022, 2023).

2. How were soil hydraulic properties aggregated from 1 km to 5 km?

As described in Section 3.1, soil parameters were regridded to the model resolution (i.e., from 1 km to 5 km) using a first-order conservative remapping, which corresponds to an area-weighted average of the soil properties.

3. Streamflow and evapotranspiration processes have distinct time scales. Are model results of the same variable at different temporal resolution, for example, daily vs. subdaily stream flow, sensitive to different model parameters?

Parameter sensitivities were exclusively analyzed with respect to the NSE of daily streamflow and the NSE of monthly evaporation as these were the objective functions that were used during calibration. In this sense, the Regional Sensitivity Analysis was performed as a previous and necessary step to the calibration that allowed for identifying the most important parameter with respect to both metrics, which were later selected as the calibration parameters. Although the effect of the temporal resolution on parameter sensitivities has not been explored in this work, it is important to mention that parameter sensitivities may indeed vary depending on the temporal resolution of the variable analyzed. For instance, Melsen and Guse (2019) showed in a large-sample application of VIC over the CONUS domain that the effect of $LAI_f$ and $rmin_f$ is negligible for the NSE of daily streamflow but becomes more important at annual scale.

4. Does calibration with only streamflow, only evapotranspiration, and both of them combined result in significantly different model parameterisations?

Please note that only two calibration experiments were performed in this study: a calibration using streamflow data (Q-only calibration) and a calibration combining streamflow and evaporation data (Q-E calibration). No calibration was performed for evaporation exclusively. This is an interesting question and is closely connected to another question posed by Anonymous Referee #1. One of the main advantages of using multiple datasets for model calibration is to reduce equifinality. This implies a decrease in the

number of behavioural parameter combinations as two or more datasets are used to constrain the model. As part of our response, we created a new figure that will be introduced in Section 4.1 and will be discussed in Section 5.1 in the revised version of the manuscript. Please refer to our response to Anonymous Referee #1 regarding the benefit of including evaporation data to reduce equifinality.

5. Using the model results in this study, can the relations between the Nash-Sutcliffe efficiencies and model parameters reported in Fig. 6 be interpreted from a hydrological point of view?

Yes, the correlations reported in Fig. 6 can be interpreted from a hydrologic perspective and it will be done in the revised version of the manuscript. This same question was posed by Anonymous Referee #1 and was extensively discussed in our answer, so please refer to our response to Anonymous Referee #1 for further details. We are convinced that the discussion on the mechanisms behind the correlations will greatly contribute to improve the manuscript and will be integrated in Section 5.1 in the revised version of the manuscript.

6. On page 11, it is reported that a simultaneous calibration with both streamflow and evapotranspiration results in a degradation of model performance. Hydrological models that have been calibrated against more than one type of data often display a greater generalisation capability to changing climate conditions. Can this be seen for the VIC model in the simulated time frame in this study? Is this what is discussed in the last paragraph of Sec. 5.2?

Please note that the deterioration of model performance that is reported on page 11 only refers to streamflow, as the performance for evaporation highly increases when the model is simultaneously calibrated with both streamflow and evaporation data (Fig. 8, 9). The generalization capabilities of the VIC model to changing climate conditions have been assessed by comparing the model performance for streamflow and evaporation for the calibration and the evaluation periods during the Split-Sample Test (Fig. 9). However, the loss in model performance during the evaluation period when compared to the calibration period was similar for both the Q-only and the Q-E calibration experiments, and therefore this effect was not visible based on the results of this study. This could be better explored by implementing a Differential Split-Sample Test (Klemeš, 1986) after selecting two contrasting periods as in Fowler et al. (2018). We acknowledge this is an interesting approach and a potential future development to further test the predictability of the VIC model. The last paragraph of Sec. 5.2 refers to the cross-validation test performed using different observational datasets of precipitation and temperature for the study period as a way to assess the generalizability of the calibrated parameters.

Fowler, K., Coxon, G., Freer, J., Peel, M., Wagener, T., Western, A., Woods, R., & Zhang, L. (2018). Simulating Runoff Under Changing Climatic Conditions: A Framework for Model Improvement. Water Resources Research, 54(12), 9812–9832. https://doi.org/10.1029/2018WR023989

7. Sec. 3.3: What climate forcing was used to spin-up the simulation? From the corresponding 10 years preceding the simulation period?

We used daily precipitation and temperature data for the 10 years preceding the simulation period (i.e., spin-up period).

---

## Author Comment (AC3)

**Responses to Anonymous Referee #3**

Dear Authors,

Thank you very much for your work. I think the work is interesting but I have some concerns.

We thank the reviewer for his/her suggestions as they will contribute to improve the manuscript. We have indicated in our responses those references that were not included in the initial version of the manuscript.

Introduction

- Introduction miss literature review: a) what about the intercomparison project MOPEX; b) what about prediction in ungauged catchments. Please, see key references, also for work that has been performed in Spain.

- Line 31: "across climates" I suggest you to have a look at Addor et al. (2018).
- Line: 40 "evaluation and benchmarking". I suggest you to have a look at Prieto et al. (2021, 2022).
- Line 43: parameter regionalization techniques, I recommend you to have a look at Almeida et al. (2016) and Prieto et al. (2019).

Addor, N., Nearing, G., Prieto, C., Newman, A. J., Le Vine, N., & Clark, M. P. (2018). A ranking of hydrological signatures based on their predictability in space. Water Resources Research, 54, 8792–8812. https://doi.org/10.1029/2018WR022606

Almeida, S., Le Vine, N., McIntyre, N., Wagener, T., and Buytaert, W. (2016). Accounting for dependencies in regionalized signatures for predictions in ungauged catchments, Hydrol. Earth Syst. Sci., 20, 887–901, https://doi.org/10.5194/hess-20-887-2016

Prieto, C., Le Vine, N., Kavetski, D., Fenicia, F., Scheidegger, A., & Vitolo, C. (2022). An exploration of Bayesian identification of dominant hydrological mechanisms in ungauged catchments. Water Resources Research, 58, e2021WR030705. https://doi.org/10.1029/2021WR030705

Prieto, C., Kavetski, D., Le Vine, N., Álvarez, C., & Medina, R. (2021). Identification of dominant hydrological mechanisms using Bayesian inference, multiple statistical hypothesis testing, and flexible models. Water Resources Research, 57, e2020WR028338. https://doi.org/10.1029/2020WR028338

Prieto, C., Le Vine, N., Kavetski, D., García, E., & Medina, R. (2019). Flow prediction in ungauged catchments using probabilistic random forests regionalization and new statistical adequacy tests. Water Resources Research, 55, 4364–4392. https://doi.org/10.1029/2018WR023254

We thank the reviewer for suggesting key references to improve the introduction. All of the references provided above will be incorporated into the introduction of the revised version of the manuscript. In addition, the MOPEX intercomparison project as well as Prediction in Ungauged Basins will be discussed from a large-sample hydrology perspective to improve the introduction.

- Line 64: "there is an increasing tendency towards aridity conditions": what is the difference for different catchments in Spain.

The increasing tendency towards aridity conditions manifests a similar pattern for the Spanish catchments and generally reveals a clear latitudinal gradient with greater aridity corresponding to the southern catchments. This will be specified in the revised version of the manuscript.

Study area and data

- Line 80: specify the northern districts (there are "many", eg, Aguas de Galicia, CHC, URA, ARA)

We will specify the Northern Districts in the text of the revised version of the manuscript. We would like to clarify that these districts were grouped under the term "Northern Districts" because the identifiers in the SAIH-ROEA dataset corresponding to all the northern catchments share the first digit.

- Also, in section 2 I recommend to provide the range of mean annual precipitation, mean annual flow, mean annual potential evapotranspiration and rainfall runoff coefficient across catchments and maybe per river basin district in the text. This is to guide the reader.

We thank the reviewer for pointing this out. We will provided the four ranges across catchments in the text to guide the reader in the revised version of the manuscript. Providing them per River Basin District can potentially confuse the reader as there are eight River Basin Districts and this would lead to a total of 32 values. The reader can always refer to Fig. 1 and Fig. 3 to visualize the hydrologic variability in space for the study catchments.

- Line 205: you are using SIMPA as benchmark which we know is a very simple model. Maybe, include the pros and cons or similarities and differences as most of the readers won't know what SIMPA is.

The importance of SIMPA for water resources management in Spain must be recognized as it constitutes a reference tool for water resources allocation at the national and at the basin scale. SIMPA is a semidistributed implementation of the lumped conceptual model of Témez (1977) and has evolved since its inception to include, among other features, a snow module (https://www.miteco.gob.es/content/dam/miteco/es/agua/temas/evaluacion-de-los-recursos-hidricos/cedex-informeerh2019_tcm30-518171.pdf), and recently a new hydrogeological module (http://hdl.handle.net/10261/335461). In our understanding, the simplicity of the original model of Témez has been already left behind, and SIMPA has been used in many previous studies for comparison purposes (e.g., Pellicer-Martínez and Martínez-Paz, 2018; Suárez-Almiñana et al., 2020; Yeste et al., 2020). We agree that describing the similarities and differences between VIC and SIMPA is going to be helpful

for the reader and beneficial for the manuscript. This description will be included in Section 3.3 in the revised version of the manuscript.

Témez, J.R., 1977. Modelo matemático de transformación "precipitación-aportación". Asociación de Investigación Industrial Eléctrica (ASINEL). Madrid.

Discussion

- I miss 1) talking about uncertainty, 2) talking about model structure error, and 3) talking or comparing (which would go to the methods) with a more well established model, e.g. GR4F, even if SIMPA is used as benchmark. So that you would have SIMPA, VIC and GR4J.

We thank the reviewer for his/her suggestions to improve the discussion of the manuscript. As per the requirement of the Anonymous Referee #1, the value of the evaporation dataset in reducing equifinality and uncertainty will be addressed according to the results of the Monte Carlo experiment. The equifinality assessment has resulted in a new figure that will be introduced in Section 4.1 and that will be discussed in Section 5.1 in the revised version of the manuscript. Please refer to our response to the major suggestion from Anonymous Referee #1 for a thorough explanation of our new findings.

In relation to model structure errors, it is not possible to analyze them when only one model structure is used as in this work. The focus of this study is the VIC model and its application for the Spanish catchments after the previous experience using VIC in Yeste et al. (2020, 2021, 2023), given the suitability of VIC for large-sample and large-scale applications (e.g., Sepúlveda et al., 2022). The evaluation of model structure errors is better performed with frameworks such as FUSE (Clark et al., 2008) or SUMMA (Clark et al., 2015). The inherent limitations of using only one model structure will be recognized in the revised version of the manuscript and the use of FUSE and SUMMA will be identified as a potential future work.

Finally, as for the comparison of VIC with a more established model such as GR4J, to our knowledge there are neither lumped GR4J simulations available for the 189 study catchments nor gridded simulations available for the Spanish domain, whereas gridded SIMPA simulations are publicly available and regularly updated. Implementing GR4J ourselves would notably require an additional calibration effort and an intensive data processing step for all the 189 study catchments to subsequently compare it against VIC. Such an endeavor is unfortunately unfeasible at this stage of project implementation and thus it is out of the scope of this work. Please see also our response to the following question.

Clark, M. P., Slater, A. G., Rupp, D. E., Woods, R. A., Vrugt, J. A., Gupta, H. V., Wagener, T., & Hay, L. E. (2008). Framework for Understanding Structural Errors (FUSE): A modular framework to diagnose differences between hydrological models. Water Resources Research, 44(12), 1–14. https://doi.org/10.1029/2007wr006735

Clark, M. P., Nijssen, B., Lundquist, J. D., Kavetski, D., Rupp, D. E., Woods, R. A., Freer, J. E., Gutmann, E. D., Wood, A. W., Brekke, L. D., Arnold, J. R., Gochis, D. J., & Rasmussen, R. M. (2015). A unified approach for process-based hydrologic modeling: 1. Modeling

concept. Water Resources Research, 51(4), 2498–2514. https://doi.org/10.1002/2015WR017198

- I also miss to compare with other models and results that were run at daily time scale in Spain, e.g. look at URA

Several studies were already discussed in Section 5.2, in particular those involving the Duero River Basin (Morán-Tejeda et al., 2014; Yeste et al., 2020, 2023), Tajo (Pellicer-Martínez and Martínez-Paz, 2018; Pellicer-Martínez et al., 2021), Guadalquivir (Yeste et al., 2018), Segura (Pellicer-Martinez and Martínez-Paz, 2015; Pellicer-Martínez et al., 2015) and Júcar (Marcos-Garcia et al., 2017; Suárez-Almiñana et al., 2020). We are happy to also include the key references suggested by the reviewer at the beginning of the review report in relation to models that were run at daily time scale for the comparison with other models and results that were run at daily time scale in northern Spain.

Conclusions

- "The soil and routing parameters were reveled as the most important parameters". Could you add in which type of catchments were most and least important?

A similar question was posed by Anonymous Referee #1 and Ilhan Özgen-Xian. Both reviewers suggested to interpret the correlations in Fig. 6 and the importance of the different parameters from a hydrologic perspective. Please refer to our response to Anonymous Referee #1 for further details, as this topic has been extensively discussed there and the discussion on the mechanisms behind the correlations will be integrated in Section 5.1 in the revised version of the manuscript. As required by the reviewer in this question, we will also indicate for which catchments the routing parameters are more important in the conclusions of the revised version of the manuscript.

Once again, thank you very much for your work.

In case the editor asks for a revised version of the manuscript, I am very happy to serve as reviewer of the revised version

All the Best,

Reviewer